# Urinary Multidrug-Resistant *Klebsiella pneumoniae*: Essential Oil Countermeasures in a One Health Case Report

**DOI:** 10.3390/microorganisms13081807

**Published:** 2025-08-01

**Authors:** Mălina-Lorena Mihu, Cristiana Ştefania Novac, Smaranda Crăciun, Nicodim Iosif Fiţ, Cosmina Maria Bouari, George Cosmin Nadăş, Sorin Răpuntean

**Affiliations:** Department of Microbiology, Immunology and Epidemiology, Faculty of Veterinary Medicine, University of Agricultural Sciences and Veterinary Medicine, 400372 Cluj-Napoca, Romania; malina.mihu@usamvcluj.ro (M.-L.M.); cristiana.novac@usamvcluj.ro (C.Ş.N.); smaranda.craciun@usamvcluj.ro (S.C.); nfit@usamvcluj.ro (N.I.F.); cosmina.bouari@usamvcluj.ro (C.M.B.); sorin.rapuntean@usamvcluj.ro (S.R.)

**Keywords:** *Klebsiella pneumoniae*, multidrug resistance, urinary tract infection, essential oils, One Health

## Abstract

Carbapenem-resistant *Klebsiella pneumoniae* (CR-Kp) is eroding therapeutic options for urinary tract infections. We isolated a multidrug-resistant strain from the urine of a chronically bacteriuric patient and confirmed its identity by Vitek-2 and MALDI-TOF MS. Initial disk-diffusion profiling against 48 antibiotics revealed susceptibility to only 5 agents. One month later, repeat testing showed that tetracycline alone remained active, highlighting the strain’s rapidly evolving resistome. Given the scarcity of drug options, we performed an “aromatogram” with seven pure essential oils, propolis, and two commercial phytotherapeutic blends. Biomicin Forte^®^ produced a 30 mm bactericidal halo, while thyme, tea tree, laurel, and palmarosa oils yielded clear inhibition zones of 11–22 mm. These in vitro data demonstrate that carefully selected plant-derived products can target CR-Kp where conventional antibiotics fail. Integrating aromatogram results into One Health’s stewardship plans may therefore help preserve last-line antibiotics and provide adjunctive options for persistent urinary infections.

## 1. Introduction

*Klebsiella pneumoniae* is increasingly recognized as a major cause of urinary tract infections (UTIs), both in community and hospital settings. As one of the foremost agents of nosocomial infections worldwide, this pathogen represents a critical One Health concern, bridging human, animal, and environmental reservoirs [1,2].

*K. pneumoniae* is ubiquitous, often found in soil, water, sewage, and vegetation, as well as within the gastrointestinal tracts of humans and animals, facilitating zoonotic transmission and environmental persistence [3]. In hospital environments, it frequently colonizes the urinary and respiratory tracts, particularly following antibiotic exposure [4]. In companion animals, particularly dogs and cats, *K. pneumoniae* has been isolated from urinary tract infections, with strains carrying resistance and virulence genes mirroring those of human isolates [5].

Clinically, *K. pneumoniae* poses a significant threat due to its ability to cause difficult-to-treat UTIs. It is one of the most frequent uropathogens associated with the rise in antibiotic resistance, particularly through the production of carbapenemases such as KPC (*K. pneumoniae* carbapenemase), which severely limits therapeutic options [6,7,8,9]. The increasing frequency, clinical burden, and treatment failure in *K. pneumoniae* UTIs highlight the urgent need for novel interventions [10,11,12].

The organism’s pathogenicity is enhanced by its arsenal of virulence factors, including a polysaccharide capsule, lipopolysaccharides (LPSs), type 1 and 3 fimbriae, and siderophores. These elements support immune evasion, promote biofilm formation, and allow adherence to uroepithelial cells, contributing to chronic and recurrent UTIs [7,13,14]. In particular, fimbriae are critical for urothelial attachment, preventing bacterial clearance during urination [13].

Of grave concern is *K. pneumoniae*’s remarkable capacity to acquire and disseminate antimicrobial resistance. Multidrug-resistant (MDR) and extensively drug-resistant (XDR) clones are spreading globally, resulting in higher morbidity, mortality, and healthcare costs [4,15]. The pathogen’s large accessory genome (~30,000 protein-coding genes) facilitates the convergence of resistance and virulence loci, potentially leading to untreatable, invasive infections [1].

Hypervirulent *K. pneumoniae* (hvKP) strains, once confined to the Asia–Pacific region, are now emerging worldwide and increasingly associated with antibiotic resistance [16,17,18]. These strains are genetically distinct from classical *K. pneumoniae* (cKP), exhibit a hypermucoviscous phenotype, and are more likely to cause disseminated, life-threatening infections [13,16,19,20,21,22]. Alarmingly, hvKP clones such as ST23 have now been documented in all six WHO regions [4], and their simultaneous acquisition of resistance traits elevates the risk of global public health emergencies [2,7,23,24].

Laboratory identification of hvKP strains can be aided by the string test for hypermucoviscosity or molecular assays targeting virulence genes [17,19,22]. The presence of these traits in urinary isolates is particularly concerning, as hvKP is capable of producing severe UTIs and pyelonephritis with systemic involvement [19].

Given that urinary tract infections are among the most prevalent bacterial infections globally, the emergence of multidrug-resistant and hypervirulent *K. pneumoniae* in this niche represents a critical healthcare challenge [6,7,9]. Nosocomial outbreaks with a wide geographic distribution and limited treatment options have already been attributed to such strains [24,25].

In response to this growing threat, the present study aimed to (i) identify and characterize a *K. pneumoniae* strain isolated from the urine of a human patient, (ii) describe its antimicrobial resistance mechanisms, and (iii) evaluate the in vitro activity of 56 antibiotics, six essential oils, a propolis tincture, and two commercial phytotherapeutic blends as potential alternatives or adjuncts in managing urinary tract infections.

## 2. Materials and Methods

### 2.1. Isolation of the Strain

The strain was isolated from an 84-year-old male patient with a history of bacteriuria. The *K. pneumoniae* isolate was obtained by streaking routine media—nutrient broth and nutrient agar—together with selective media (MacConkey agar and UriSelect^TM^, Bio-Rad Laboratories Inc., Hercules, CA, USA) that facilitate the preliminary recognition of *Enterobacteriaceae*. After inoculation, tubes and Petri plates were aerobically incubated at 37 °C in a thermostatic chamber and examined after 24 and 48 h.

### 2.2. Definitive Identification

Species confirmation was achieved with two automated systems, Vitek^®^ 2 Compact 15 (bioMérieux, Craponne, France), which uses an array of biochemical reactions to generate species-level identifications, and MALDI Biotyper^®^ Sirius System (Bruker, Ettlingen, Germany), which applies matrix-assisted laser desorption/ionization time-of-flight mass spectrometry (MALDI-TOF MS) to analyze the bacterial protein profile (CLSI, 2023 [26]).

### 2.3. Antimicrobial Susceptibility Testing

Susceptibility was evaluated by the standard Kirby–Bauer agar-disk diffusion method, interpreted according to EUCAST breakpoints but cross-checked with CLSI zone–diameter tables. A 0.5 McFarland suspension of the isolate was swab-spread onto Mueller–Hinton agar (Merck, Darmstadt, Germany) plates; antibiotic disks (commercially coded micro-tablets) were then placed on the surface. Plates were incubated at 35 ± 1 °C for 18 ± 2 h. Inhibition zones were measured to the nearest millimeter with a digital vernier caliper and categorized as susceptible (S), intermediate (I), or resistant (R). Two evaluations were performed using the same antibiotics at one month apart. For the second susceptibility test, *K. pneumoniae* was re-isolated from a fresh urine sample collected from the same patient. The strain was identified again using the same microbiological and biochemical methods as in the initial testing phase in order to confirm its identity prior to further antimicrobial susceptibility assessments.

The tested antimicrobials were as follows (drug/code/content): mecillinam (MEC, 10 µg), piperacillin (PRL, 30 µg), amoxicillin + clavulanic acid (AMC, 20/10), cefadroxil (CFR, 30 µg), cefaclor (CEC, 30 µg), cefotetan (CTT, 30 µg), cefuroxime (CXM, 30 µg), cefamandole (MA, 30 µg), ceftriaxone (CRO, 30 µg), ceftazidime (CAZ, 10 µg), ceftazidime + avibactam (CZA, 10/4 µg), cefoperazone (CEP, 75 µg), ceftiofur (FUR, 30 µg), cefixime (CFM, 5 µg), cefquinome (CFQ, 30 µg), meropenem (MEM, 10 µg), imipenem (IMI, 10 µg), amikacin (AK, 30 µg), streptomycin (S, 10 µg), gentamicin (GME, 10 µg), azithromycin (AZM, 15 µg), tetracycline (TE, 30 µg), doxycycline (DOX, 30 µg), tulathromycin (TUL, 30 µg), norfloxacin (NX, 10 µg), marbofloxacin (MAR 5 µg), ofloxacin (OFX, 5 µg), chloramphenicol (C, 30 µg), nitrofurantoin (F, 100 µg), and trimethoprim + sulfamethoxazole (SXT, 1.25/23.75 µg). All commercial antibiotic disks were purchased from Liofilchem^®^, Roseto degli Abruzzi, Italy, and Bio-Rad Laboratories Inc., Hercules, CA, USA.

### 2.4. Essential Oil and Antiseptic Susceptibility Testing

Sterile blank filter paper disks (⌀ 6 mm) were impregnated with 10 µL of each test solution—neat essential oil, tincture, or antiseptic—then placed on Mueller–Hinton agar plates that had been lawn-inoculated with the *K. pneumoniae* 0.5 McFarland suspension, following the same layout as the antibiotic disk-diffusion assay. Plates were incubated at 37 °C for 24 h and re-read at 48 h; inhibition halos were measured in millimeters. Complete absence of growth within the halo was interpreted as bactericidal activity, whereas satellite colonies were recorded as evidence of partial resistance. Essential oils screened were represented by palmarosa (*Cymbopogon martini*), geranium (*Pelargonium graveolens*), frankincense (*Boswellia carteri*), laurel (*Laurus nobilis*), tea tree (*Melaleuca alternifolia*), and thyme (*Thymus vulgaris*), all purchased from Elemental SRL, Oradea, Romania. Natural remedy formulations were represented by propolis tincture, Biomicin Urinar^®^ (A20, Fares, Orăștie, Romania), and Biomicin Forte^®^ (A3, Fares, Orăștie, Romania). An antiseptic comparator represented by a 1% methylene blue solution was also used. Moreover, a standard antibiotic disk containing amoxicillin + clavulanic acid (AMC) was included on each plate as a broad-spectrum reference control.

## 3. Results

### 3.1. Cultural Characterization

Regarding growth in the nutrient broth, after 24 h of incubation, the broth became markedly turbid, with a thick surface pellicle that later sedimented to the bottom of the tube. When the tubes were tilted or gently rotated, the culture adhered to the glass walls, indicating abundant extracellular mucus. With aging (48–72 h), the broth’s viscosity increased further because of copious capsular-mucus production. The hyper-mucoviscous phenotype was most obvious when the culture was withdrawn with a loop or Pasteur pipette, producing long, filamentous strings (“string test” positive).

After 24 h on solid media, the isolate formed colonies whose size, surface appearance, pigmentation, and consistency evolved over time (48–72 h) as follows: On nutrient and Mueller–Hinton agar: large (3–5 mm diameter), convex, opaque, glossy, and non-pigmented colonies (Appendix A). On UriSelect^TM^ chromogenic agar, colonies were similar in size and shape, but purple in color, consistent with the primary differentiation of *Enterobacteriaceae*. On blood agar, intensely shiny, non-hemolytic colonies that clearly exhibited the mucoid character were observed. Upon prolonged incubation, the colonies enlarged, acquired a faint pink hue, and tended to coalesce (Appendix A).

### 3.2. Antibiotic Susceptibility Testing

Regarding the first evaluation, out of the 30 antibiotics and chemotherapeutics agents tested, susceptibility was demonstrated for only 5 (16.67%, CI 95% 5.64–34.72) as follows: streptomycin, tetracycline, doxycycline, chloramphenicol, and tulathromycin. Resistance was recorded in a large number of the agents tested, 25 in total (83.33%, CI 95% 65.28–94.36) (Table 1).

On the other hand, the second evaluation revealed a higher resistance rate, with only one efficient antibiotic (3.33%, CI 95% 0.08–17.22), namely tetracycline, and the bacterial strain being resistant to 29 antimicrobials (96.67%, CI 95% 82.78–99.92). The results are presented in Table 1.

Serial testing revealed a precipitous loss of activity, so streptomycin, tulathromycin, chloramphenicol, and doxycycline all flipped from susceptible to resistant between evaluations, leaving tetracycline as the sole agent retaining efficacy (Figure 1).

### 3.3. Essential Oil Susceptibility Testing

The antimicrobial activity of the tested essential oils and natural products against *K. pneumoniae* varied significantly. The largest inhibition zones were observed for Biomicin Forte^®^ (A3) and *Thymus vulgaris* (thyme), with diameters of 30 mm and 22 mm, respectively, indicating strong antibacterial properties. *Melaleuca alternifolia* (tea tree oil) showed moderate efficacy, with inhibition zones of 20 mm. Moderate activity was also recorded for *Laurus nobilis* (laurel) and Biomicin Urinar^®^ (A20), both yielding zones around 12 mm, although the latter showed the presence of resistant colonies. In contrast, several oils, including *Pelargonium graveolens* (geranium), *Boswellia carteri* (frankincense), *Cymbopogon nardus* (citronella), and propolis, exhibited no inhibitory activity, as indicated by resistant growth. These findings highlight considerable variability in the susceptibility of *K. pneumoniae* to different natural products, suggesting potential for selective use in antimicrobial strategies. The results are presented in Table 2.

## 4. Discussion

*Klebsiella pneumoniae* remains a leading cause of healthcare-associated infections worldwide, driven by a highly diverse population structure that complicates both genomic surveillance and clinical management. Kleborate, a recently developed analytic pipeline, now streamlines genotype to phenotype prediction directly from intestinal metagenomes and cultured isolates, offering a practical answer to this complexity [27]. MDR in *K. pneumoniae* arises through four, often co-occurring, mechanisms: (i) production of extended-spectrum β-lactamases (ESBLs), (ii) decreased outer-membrane permeability via porin loss (e.g., OmpK35/OmpK36), (iii) over-expression of efflux pumps such as the intrinsic MFS pump KpnGH, and (iv) modification of antimicrobial targets [24,28]. Most resistance determinants reside on mobile genetic elements such as plasmids, transposons, and integrons, facilitating both vertical inheritance and horizontal exchange within and across species [2,29].

Plasmids are typically circular, autonomously replicating DNA molecules, although linear variants have been documented. Standardized multiplex PCR enables rapid plasmid typing, a prerequisite for tracking the dissemination of drug-resistance cassettes in *K. pneumoniae* populations [30,31]. Conjugative plasmids, in particular, encode the full complement of transfer machinery and therefore mediate inter-strain spread over large taxonomic distances [32]. Specific β-lactamase and carbapenemase genes often segregate with discrete plasmid backbones, and copy number amplification can further boost resistance levels. Porin loss, particularly of OmpK36, synergizes with plasmid-borne ESBLs and carbapenemases to confer pan-β-lactam resistance [33].

MDR *K. pneumoniae* lineages are genomically plastic, engaging in frequent chromosomal recombination that reshuffles the capsule (K-locus) and O-antigen loci [34]. By contrast, hyper-virulent clones tend to recombine less but acquire large virulence plasmids, siderophore systems, and regulators of the hyper-mucoid phenotype [35,36]. Alarmingly, recent reports describe pathotypes that unite extensive drug resistance with hyper-virulence, creating both untreatable and highly invasive strains [25,37]. Global travel and healthcare tourism accelerate their dispersal, highlighting the need for continuous genomic surveillance.

Our isolate displayed broad resistance encompassing aminoglycosides, β-lactams (penicillins, cephalosporins, and carbapenems), macrolides, fluoroquinolones, and several second-line classes. Serial susceptibility testing revealed a precipitous loss of activity: streptomycin, tulathromycin, chloramphenicol, and doxycycline transitioned from susceptible to resistant between two evaluations, leaving tetracycline as the sole agent retaining efficacy. The phenotype aligns with the presence of transferable ESBL/carbapenemase genes combined with porin down-regulation and active efflux.

Concerning a class-by-class overview, among aminoglycosides, only streptomycin exhibited initial activity, consistent with its distinct binding affinity for the 30S ribosomal subunit [29]. In contrast, complete resistance to penicillins and other β-lactams was observed, most likely due to high-level ESBL and carbapenemase production [29,38,39,40,41]. Within the macrolide class, intrinsic resistance was predominant; however, the veterinary compound tulathromycin showed transient efficacy. Active efflux mechanisms are thought to contribute significantly to this resistance phenotype [29]. Regarding other antibiotic classes, resistance to fluoroquinolones, sulfonamides, and nitrofuran derivatives was universal. Tetracycline and chloramphenicol displayed some measurable inhibitory activity, but both agents were quickly rendered ineffective by the rapid emergence of resistance.

The worldwide rise in microbial resistance to conventional chemicals and drugs has spurred intensive searches for new broad-spectrum biocides and alternative therapeutic strategies [42,43]. Essential oils (EOs) exhibit significant antibacterial activity by disrupting the bacterial cell membrane, increasing its permeability, and causing the leakage of vital intracellular components, which ultimately leads to cell death [44]. Moreover, pathogens appear unable to develop resistance to EOs because the oils contain such a wide variety of components that adaptive mutation is virtually impossible.

Natural products such as EOs are promising because of their complex composition; they have already proved effective against drug-resistant *K. pneumoniae* strains, although their overall mechanisms are not yet fully elucidated [45,46]. EOs contain diverse secondary metabolites capable of inhibiting or slowing the growth of bacteria, yeasts, and molds, mainly by targeting the cell membrane and cytoplasm and, in some cases, profoundly altering cell morphology [42]. Plant-derived EOs from oregano (*Origanum vulgare*), sage (*Salvia officinalis*), and thyme (*Thymus vulgaris*) have attracted attention because they may substitute for failing traditional antibiotics against pathogens, including *Klebsiella* spp. [47].

The increasing antimicrobial resistance of *Klebsiella pneumoniae* has prompted the need for alternative approaches, particularly in infections where conventional antibiotics have limited efficacy. In our study, two consecutive susceptibility tests performed on the same *K. pneumoniae* isolate (one month apart) revealed a consistent resistance pattern to most conventional antibiotics. Notably, tetracycline remained the only antibiotic to which the isolate was susceptible in both tests, suggesting its potential retained efficacy against this strain.

Given the reduced effectiveness of conventional drugs, we assessed the antibacterial activity of selected natural products and essential oils. Among the tested alternatives, Biomicin Forte, thyme essential oil, and tea tree essential oil demonstrated the largest inhibition zones, indicating strong in vitro efficacy against the multidrug-resistant *K. pneumoniae* strain. These results align with previous reports highlighting the antimicrobial properties of thymol and terpinen-4-ol, the main active components in thyme and tea tree oils, respectively [48,49].

Propolis, palmarosa, geranium, laurel, citronella, and frankincense oils also showed variable antibacterial effects, although to a lesser extent. Interestingly, Biomicin Urinar^®^, a natural antimicrobial mixture formulated for urinary tract infections, exhibited moderate activity, supporting its potential clinical relevance.

The differential efficacy among the tested products may be attributed to their diverse phytochemical profiles, particularly in terms of phenolic compounds and terpenoids known to affect bacterial membrane integrity and metabolic functions [44]. Furthermore, the complex composition of essential oils is thought to reduce the likelihood of resistance development, a hypothesis supported by the consistent results observed between the two testing time points [50].

While our in vitro results are promising, further studies are needed to evaluate the pharmacokinetics, safety, and clinical effectiveness of these natural antimicrobials, especially regarding their comparison and synergy with conventional antibiotics. Nevertheless, our findings support the integration of selected essential oils and natural products—particularly Biomicin Forte^®^, thyme, and tea tree oils—into the pipeline of potential adjunct therapies against multidrug-resistant *K. pneumoniae*.

The convergence of multidrug resistance and hyper-virulence in *K. pneumoniae* heightens the urgency for innovative countermeasures. Continuous genomic surveillance, strict antimicrobial stewardship, and the development of adjunctive therapies, ranging from monoclonal antibodies to tailored essential oil formulations, are pivotal to preserving treatment efficacy and safeguarding public health.

## 5. Conclusions

Essential oil screening revealed consistent inhibitory activity against the isolated *Klebsiella pneumoniae* strain for palmarosa (*Cymbopogon martini*), laurel (*Laurus nobilis*), tea tree (*Melaleuca alternifolia*), and thyme (*Thymus vulgaris*) oils. In contrast, geranium, frankincense, citronella oil, and propolis tincture showed no measurable activity. Among the tested commercial products, Biomicin Urinar^®^ and especially Biomicin Forte^®^ produced stable and well-defined inhibition zones, with Biomicin Forte^®^ demonstrating a bactericidal effect. The absence of resistant subcolonies within these halos suggests that selected essential oils may hold potential as adjunctive therapeutic agents for *K. pneumoniae* urinary infections, provided that their selection is guided by a laboratory “aromatogram” performed alongside the conventional antibiogram.

## Figures and Tables

**Figure 1 microorganisms-13-01807-f001:**
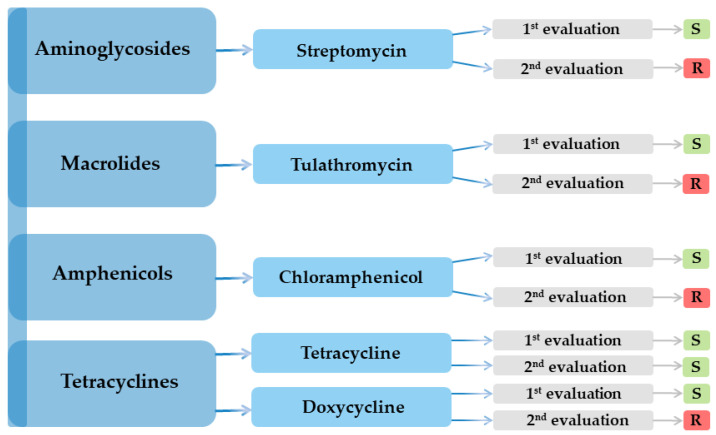
The *K. pneumoniae* strain’s antibiotic susceptibility variation.

**Table 1 microorganisms-13-01807-t001:** Antibiotic susceptibility profile of the *K. pneumoniae* isolate.

Antimicrobial Class	Nr.	Antibiotic	Result	
			1st Evaluation	2nd Evaluation
Aminoglycosides	1	Amikacin (AK)	R	R
	2	Streptomycin (S)	S	R
	3	Gentamicin (GME)	R	R
Penicillins	4	Piperacillin (PRL)	R	R
	5	Mecillinam (MEC)	R	R
	6	Amoxicillin–clavulanic acid (AMC)	R	R
Macrolides	7	Azithromycin (AZM)	R	R
	8	Tulathromycin (TUL)	S	R
Cephalosporins 1st generation	9	Cefadroxil (CFR)	R	R
2nd generation	10	Cefaclor (CEC)	R	R
	11	Cefotetan (CTT)	R	R
	12	Cefuroxime (CXM)	R	R
	13	Cefamandole (MA)	R	R
3rd generation	14	Ceftriaxone (CRO)	R	R
	15	Ceftazidime (CAZ)	R	R
	16	Cefoperazone (CEP)	R	R
	17	Ceftiofur (FUR)	R	R
	18	Cefixime (CFM)	R	R
	19	Ceftazidime-avibactam (CZA)	R	R
4th generation	20	Cefquinome (CFQ)	R	R
Carbapenems	21	Meropenem (MEM)	R	R
	22	Imipenem (IMI)	R	R
Fluoroquinolones	23	Norfloxacin (NX)	R	R
	24	Ofloxacin (OFX)	R	R
	25	Marbofloxacin (MAR)	R	R
Tetracyclines	26	Tetracycline (TET)	S	S
	27	Doxycycline (DOX)	S	R
Amphenicols	28	Chloramphenicol (C)	S	R
Chemotherapeutic agents	29	Nitrofurantoin (F)	R	R
Sulfonamides	30	Trimethoprim–sulfamethoxazole (SXT)	R	R

R—resistant; S—susceptible.

**Table 2 microorganisms-13-01807-t002:** Susceptibility/resistance of the *K. pneumoniae* isolate to essential oils.

International Name	Latin Name/Composition	Inhibition Area Diameter (mm)
Palmarosa	*Cymbopogon martini*	11
Geranium	*Pelargonium graveolens*	R
Frankincense	*Boswellia carteri*	R
Laurel	*Laurus nobilis*	12
Tea tree	*Melaleluca alternifolia*	20
Citronella	*Cymbopogon nardus*	R
Thyme	*Thymus vulgaris*	22
Propolis	*Apis mellifera* propolis	R
Biomicin Urinar^®^ (A20)	*Origani aetheroleum* + *Cinnamomun verum* + *Salvia officinalis* + *Thymi aetheroleum*	12 (resistant colonies)
Biomicin Forte^®^ (A3)	*Thymi aetheroleum* + *Caryophylli floris aetheroleum*	30
Methylene blue 3%	*-*	13

R—resistant.

## Data Availability

The original contributions presented in this study are included in the article/Appendix A. Further inquiries can be directed to the corresponding author.

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
