# Peer review of "Urinary Multidrug-Resistant Klebsiella pneumoniae: Essential Oil Countermeasures in a One Health Case Report"

_microorganisms, 2025, doi:10.3390/microorganisms13081807_

Round 1
Reviewer 1 Report
Comments and Suggestions for Authors
Although the article addresses a timely and relevant topic and is generally well-structured and conceived, it nonetheless requires substantial improvement to meet the standards expected for publication:
1. The Introduction should be more concise, avoiding the inclusion of information that is not directly relevant to the main topic of the article, in order to enhance the clarity and precision of this section.
2. In the initial methodology for the isolation of the Klebsiella strain from urine, only the selective media used, relevant for the identification of this bacterial species, should be mentioned.
3. Subsection 2.2 does not significantly contribute to the objectives of the study, and the information presented therein can be omitted without affecting the overall coherence of the article.
4. The selection of antibiotics for antimicrobial susceptibility testing should be aligned with the recommendations of the CLSI and EUCAST guidelines, which clearly specify the antibiotics that are appropriate for testing Klebsiella strains. Reporting results for antibiotics to which Klebsiella is naturally resistant is methodologically unjustified.
5. It is necessary to indicate the criteria used for interpreting the results of essential oil testing as susceptible (S) or resistant (R).
6. The Conclusions should focus exclusively on the effect of the essential oils on the isolated strain, without including information related to its characterization.
Author Response
Dear Reviewer 1,
Thank you very much for your thoughtful and insightful comments on our manuscript. We truly appreciate the time and expertise you invested in evaluating our work. Your constructive feedback has helped us identify several areas for improvement, and we have carefully addressed each point in the revised version. We believe these changes have strengthened both the clarity and the scientific rigor of the paper, and we are grateful for your contribution to this process.
Please find the rest of the comments in the attachment.

Reviewer 2 Report
Comments and Suggestions for Authors
This is interesting and up-to date investigation on antibiotic resistance profile of isolated MDR K. pneumonaiae and its susceptibility to alternative antimicrobials - essential oils and their commercial mixtures and propolis. However, some revision is necessary prior to acceptance.
The main points that should be revised are listed here, while all the points (the main and minor ones) are labeled directly in the manuscript:
- The authors provided information concerning (1) urine culture screening which was applied in a private laboratory during long time period (06.02.2024-25.04.2025) and (2) screening of K. pneumoniae susceptibilities to conventional antibiotics (performed succesively in one month period) and alternative antimicrobials (essential oils, oils mixtures and propolis). However, Material and Methods section contains only information on the K. pneumoniae susceptibility testing. Further on, connection between these two parts remained unclear, including testing chronology. It seems that urine culture screening was performed prior to K. pneumoniae isolation, identification and susceptibility te4sting, but that should be clearly stated. Some revision throughout the manuscript, in order to elucidate this point, is necessary.
- Some confusion was made concerning the thyme essential oil tested in this study. Under the same name Thyme, the authors have written that they tested Thymus vulgaris (within the manuscript text) and Summer savory/Satureja hortensis (within Supplementary). Further on, summer savory is a common name of a plant with Latin name Satureja hortensis. This has to be revised and clear statement on the tested substance (correctly named), should be mentioned wherever it is necessary.
- Part of discussion entitled Class-by-class overview (lines 261-275) is unclear and technically incorrectly written (in the form of items list which resembles to the draft version). Following revision of this part should be made:
(1) Classical discussion form should be provided, instead of an items list. For example: Concerning class-by class overview, it is well known that aminoglicosides act by irreversible binding to the 30S ribosomal subunit. Among aminoglycosides, str was initially active. Further, beta-lactams....etc.
(2) Instead discussion of alternative therapeutics (polypeptides, monoclonal antibodies and phages), which were not screened in this research, the authors should rather focus on clear discuss of the results obtained in their manuscript (susceptibility to tested antibiotics). - Discussion on alternative antimicrobials activity (essential oils and their commercial mixtures, and propolis, lines 291-305) is written unclear and unprofessionally (items list of some constituents/essential oils activity, with unclear selection of mentioned constituents; discussion is focused on some constituents without taking into account possible synergism between them within the essential oils). This part of the discussion should also be carefully revised, with special attention paid on the most active Biomicin forte mixture and its constituents (essential oils of thyme and Caryophyllus, and their pure compounds).
- Concerning Supplementary, Petri dish 9 should be labeled.

Author Response
Dear Reviewer 2,
We want to thank you sincerely for your thoughtful, constructive, and encouraging review. The changes also considered minor comments from the document you kindly provided.
Your insights have helped us improve both the clarity and impact of the manuscript. In the attachment, we provide a point-by-point reply.

Round 2
Reviewer 1 Report
Comments and Suggestions for Authors
I would like to thank the authors for accepting the suggestions and for their thorough revision of the manuscript. The revised version shows clear improvement.
Reviewer 2 Report
Comments and Suggestions for Authors
The manuscript has been upgreaded and can be accepted in this form.